# Diffusion Curvature for Estimating Local Curvature in High Dimensional Data

**Dhananjay Bhaskar**
Department of Genetics
Yale University
dhananjay.bhaskar@yale.edu

**Kincaid MacDonald**
Department of Mathematics
Yale University
kincaid.macdonald@yale.edu

**Oluwadamilola Fasina**
Applied Mathematics Program
Yale University
dami.fasina@yale.edu

**Dawson Thomas**
Department of Mathematics
Department of Physics
Yale University
dawson.thomas@yale.edu

**Bastian Rieck**
Institute of AI for Health
Helmholtz Pioneer Campus, Helmholtz Munich
bastian.rieck@helmholtz-muenchen.de

**Ian Adelstein**
Department of Mathematics
Yale University
ian.adelstein@yale.edu

**Smita Krishnaswamy**
Department of Computer Science
Department of Genetics
Applied Mathematics Program
Program for Computational Biology and Bioinformatics
Yale University
smita.krishnaswamy@yale.edu

## Abstract

We introduce a new intrinsic measure of local curvature on point-cloud data called *diffusion curvature*. Our measure uses the framework of diffusion maps, including the data diffusion operator, to structure point cloud data and define local curvature based on the laziness of a random walk starting at a point or region of the data. We show that this laziness directly relates to volume comparison results from Riemannian geometry. We then extend this scalar curvature notion to an entire quadratic form using neural network estimations based on the diffusion map of point-cloud data. We show applications of both estimations on toy data, single-cell data and on estimating local Hessian matrices of neural network loss landscapes.

## 1 Introduction

With the advent of high-throughput high-dimensional point-cloud data in many fields including biomedicine, social science, physics, and finance, there is an increasing need for methods to extract structure and patterns. A key framework for exploring such data is the manifold hypothesis, which states that high-dimensional data arise as samples from an underlying lower dimensional manifold. The diffusion geometry and diffusion map framework first introduced by Coifman et al. [1] has proven to be a useful framework for the shape and structure of the data. Diffusion geometry involves

36th Conference on Neural Information Processing Systems (NeurIPS 2022).

converting data into an affinity graph which is then Markov normalized to form a data diffusion operator. This diffusion operator allows for the understanding of geometric features of the data such as manifold-intrinsic distances, and intrinsic manifold eigendimensions (also known as diffusion maps). However, curvature, a salient feature of Riemannian geometry, has received less attention. Being an inherently smooth quantity, its extension to the discrete case is not straightforward. The most authoritative definitions of curvature in the discrete setting have been that of Ollivier–Ricci [2] and of Forman [3], which compute curvature on edges of a graph. By contrast, here we propose a method of estimating curvature at *points or regions* in point-cloud data, where there are no inherent edges to consider—one of the first such formulations to our knowledge.

To achieve this, we utilize the connection between data diffusion and volume to define a new formulation of curvature known as *diffusion curvature* based on the volume growth of graph diffusion starting from a vertex. Starting from a unit mass placed at one or several points, we show that in cases of positive curvature the volume growth tends to be slower with more of the mass remaining in the original point, whereas in cases of negative curvature the mass tends to disperse faster to other points. Therefore the ratio of mass in the original point, compared to the remainder of the cloud gives a scalar value of curvature. We note that this method avoids the complexity of having to consider all (potentially exponentially many) edges that would arise by simply attempting to directly translate notions of edge curvature to the data affinity graph. In addition to the scalar notion of curvature, we also show that we are able to compute a quadratic form of arbitrarily high dimensions using neural network training based on diffusion probabilities as the input.

We validate our notion of curvature on toy datasets as well as real single-cell point cloud data of very high dimensionality. Then, we study both the scalar diffusion as well as the diffusion based quadratic form in studying the area surrounding found minima in neural network loss landscapes. The Hessian of the parameter space is generally difficult to compute as it contains $\mathcal{O}(n^2)$ entries for a neural network with $n$ parameters. However, restricting ourselves to the area around a local minima yields a lower dimensional space, where only a subset of parameter combinations change in the vicinity. We sample in this lower dimensional space and compute a diffusion operator which allows us to assess the curvature of the minima. First, we can assess if the point of convergence is indeed a minimum or potentially a saddle. Further, studies have shown that flat minima, i.e., minima situated in a neighborhood of the loss landscape with roughly similar error, tend to generalize better than sharp minima. These qualities can be assessed from curvature estimations of minima regions.

Our **key contributions** include:

- The formulation of diffusion curvature, an intrinsic scalar valued curvature for regions of high dimensional data sampled from an underlying manifold, computed from data diffusion.
- Proof that our notion of curvature will result in higher values in positive curvature based on volume comparison results from Riemannian geometry.
- A neural network version that computes a quadratic form starting from a diffusion map.
- Validation of our framework on toy and single cell datasets.
- Application to Hessian estimation in neural networks.

## 2 Background

### 2.1 High Dimensional Point Clouds and the Manifold Assumption

A useful assumption in representation learning is that high-dimensional data is sampled from an intrinsic low-dimensional manifold that is mapped via nonlinear functions to observable high-dimensional measurements; this is commonly referred to as the manifold assumption. Formally, let $\mathcal{M}^d$ be a hidden $d$ dimensional manifold that is only observable via a collection of $n \gg d$ nonlinear functions $f_1, \ldots, f_n \colon \mathcal{M}^d \to \mathbb{R}$ that enable its immersion in a high-dimensional ambient space as $F(\mathcal{M}^d) = \{\mathbf{f}(z) = (f_1(z), \ldots, f_n(z))^T : z \in \mathcal{M}^d\} \subset \mathbb{R}^n$ from which data is collected. Conversely, given a dataset $X = \{x_1, \ldots, x_N\} \subset \mathbb{R}^n$ of high-dimensional observations, manifold learning methods assume data points originate from a sampling $Z = \{z_i\}_{i=1}^N \subset \mathcal{M}^d$ of the underlying manifold via $x_i = \mathbf{f}(z_i)$, and aim to learn a low dimensional intrinsic representation that approximates the manifold geometry of $\mathcal{M}^d$.

### 2.1.1 Diffusion Maps

To learn a manifold geometry from collected point cloud data, we use the popular diffusion maps construction [1]. This construction starts by considering local similarities defined via a kernel $\mathcal{K}(x, y)$, with $x, y \in F(\mathcal{M}^d)$, that captures local neighborhoods in the data. While the Gaussian kernel is a popular choice for $\mathcal{K}$, it encodes sampling density information in its computation. Hence, to construct a diffusion geometry that is robust to sampling density variations, we use an anisotropic kernel

$$\mathcal{K}(x, y) = \frac{\mathcal{G}(x, y)}{\|\mathcal{G}(x, \cdot)\|_1^\alpha \|\mathcal{G}(y, \cdot)\|_1^\alpha}$$

with $\|\cdot\|_1$ the 1-norm (or row sum) and $\mathcal{G}(x, y) = e^{-\|x - y\|^2/\sigma}$ as proposed in [1], where $0 \leq \alpha \leq 1$ controls the separation of geometry from density, with $\alpha = 0$ yielding the classic Gaussian kernel, and $\alpha = 1$ completely removing density and providing a geometric equivalent to uniform sampling of the underlying manifold. In [1], it is shown that in the limit of infinitely many points this becomes equivalent to a Laplace–Beltrami operator, which encodes geometric properties of the underlying manifold. Next, the similarities encoded by $\mathcal{K}$ are normalized to define transition probabilities $p(x, y) = \mathcal{K}(x,y)/\|\mathcal{K}(x,\cdot)\|_1$ that are organized in an $N \times N$ row stochastic matrix $\mathbf{P}_{ij} = p(x_i, x_j)$ that describes a Markovian diffusion process over the intrinsic geometry of the data. Finally, a diffusion map is defined by taking the eigenvalues $1 = \lambda_1 \geq \lambda_2 \geq \cdots \geq \lambda_N$ and (corresponding) eigenvectors $\{\phi_j\}_{j=1}^N$ of $\mathbf{P}$, and mapping each data point $x_i \in X$ to an $N$ dimensional vector:

$$\Phi_t(x_i) = [\lambda_1^t \phi_1(x_i), \ldots, \lambda_N^t \phi_N(x_i)]^T \tag{1}$$

Here $t$ represents a diffusion-time. In general, as $t$ increases, most of the eigenvalues become negligible; truncated diffusion map coordinates can thus be used for dimensionality reduction, and Euclidean distances in this space are a manifold-intrinsic distance [1, 4]. Note that the trivial eigenvector, $\phi_1$ (with corresponding eigenvalue $\lambda_1 = 1$), is often omitted since it represents the stationary distribution of the Markov random walk and does not contain information about the data. Previous work has demonstrated robustness to noise and outlined the best practices for choosing kernel parameters for various applications, including diffusion-based dimensionality reduction and manifold learning of single-cell data [5, 6, 7].

The eigenvalues of the Laplace-Beltrami operator, as well as the eigenvalues of the diffusion map, encode geometry information about a compact Riemannian manifold [1], as can be seen via the asymptotic expansion of the trace of the heat kernel [8]. Dimension, volume, and total scalar curvature are spectrally determined properties of the manifold. Given the relationship between diffusion coordinates and the eigenvectors of the Laplace-Beltrami operator, one can surmise that the diffusion operator also encodes geometric information. These relations motivate our use of diffusion to measure curvature on a data manifold. Section A5 in the Appendix contains additional details in the Riemannian setting.

### 2.2 Discrete Curvature in Riemannian Geometry

There is a long history in Riemannian geometry of using curvature to study geometric and topological properties of a Riemannian manifold. In particular, lower bounds on Ricci curvature have been related to diameter (Bonnet-Myers [9]), volume (Bishop-Gromov [10]), the Laplacian (Cheng-Yau [11]), the isoperimetric constant (Buser [12]), and topological properties of the manifold (Hamilton's Ricci flow [13]). Although somewhat paradoxical, there has recently been work to extend these smooth Riemannian ideas to the discrete setting, in particular to graphs and Markov chains. Especially notable in this regard is the definition of discrete Ricci curvature due to Ollivier which makes use of the transport distance between probability distributions.

Ollivier's Ricci curvature [2] starts with a metric space $X$ equipped with a random walk (a probability measure $m_x(\cdot)$ for each $x \in X$) and assigns the edge-wise scalar curvature $k_{OR}(x, y) = 1 - W_1(m_x,m_y)/d(x,y)$ where $W_1(\cdot, \cdot)$ is the $L^1$ transportation distance (also known as earth mover's distance or Wasserstein distance). In the Riemannian setting, if one defines the random walk to be $dm_x^r(y) = dvol(y)/volB(x,r)$ then Ollivier demonstrates that $k_{OR}(x, y) = r^2 Ric(v,v)/2d+2 + O(r^3 + d(x, y)r^2)$ where $v$ is a unit tangent vector, and $y$ is a point on the geodesic issuing from $v$ with $d(x, y)$ small enough. Ollivier (and others [14], [15]) have used lower bounds on this notion of curvature to study global properties of the space (i.e. diameter, volume growth, spectral gap).

The main downsides of Ollivier's Ricci curvature are that it is an edge-wise notion and that it requires knowledge of the transport distance. A priori point cloud data does not come with a graph structure. We therefore propose a diffusion based method for prescribing a point-wise scalar curvature to point cloud data. Like Ollivier's notion, this *diffusion curvature* is based on the idea that the spread of geodesics is influenced by Riemannian curvature. We prove theoretical bounds on the diffusion curvature by appealing to the Bishop-Gromov volume comparison theorem.

## 2.3 The Loss Landscape and Hessian of Neural Networks

In the context of neural networks, second-order information about the loss function contains useful information about the generalizability of minima, the heuristic being that *flat* minima generalize better than *sharp* minima. Previous studies have sought to directly examine the loss landscape of a neural network. Li et al. [16] used random samples around found minima to visualize and quantify sharpness of minima, Horoi et al. [17] utilized jump-and-retrain sampling to visualize and classify "good" and "bad" minima. However, computing the full Hessian of the parameter space is only possible in special cases such as variational quantum classifiers [18] and it is generally intractable for high-dimensional systems (such as the parameter space of a neural network). Consequently, previous work has sought to approximate the largest (and smallest) eigenvalues of the Hessian [19, 20], yielding an incomplete picture. To circumvent this, we estimate a low dimensional Hessian of the loss function around its optimum using a quadratic approximation. Restricting ourselves to the area around the optimum yields a lower dimensional space, where only a subset of parameter combinations change in the vicinity. This Hessian approximation is achieved by sampling around the optimum, constructing a diffusion operator based on these sampled points, and using a neural network to learn the coefficients of the quadratic approximation.

Neural networks are trained such that their output function $f(X, \theta)$ matches a given function $y(X)$ on training data $X$ by performing, e.g., stochastic gradient descent on a loss function comparing $y$ and $f$ with respect to the parameters $\theta$. A typical example of a loss function is mean squared error $\mathcal{L}(X, \theta) = \sum_{x \in X} ||f(x) - y(x)||_2$. The loss function defines a loss landscape for fixed training data $X$. Here, we use data diffusion to directly estimate the Hessian of the loss landscape, with respect to the parameters of a neural network, around found minima, providing a quantitative measure of the region in terms of its spectrum and condition number. In general, the Hessian of a scalar-valued function, $\mathcal{L}(\theta)$ is a symmetric matrix of partial derivatives with the quadratic form:

$$H\mathcal{L}(\mathbf{v}) = \begin{pmatrix} v_1 & \cdots & v_n \end{pmatrix} \begin{pmatrix} \frac{\partial^2 \mathcal{L}}{\partial \theta_1 \partial \theta_1} & \cdots & \frac{\partial^2 \mathcal{L}}{\partial \theta_1 \partial \theta_n} \\ \vdots & \ddots & \vdots \\ \frac{\partial^2 \mathcal{L}}{\partial \theta_n \partial \theta_1} & \cdots & \frac{\partial^2 \mathcal{L}}{\partial \theta_n \partial \theta_n} \end{pmatrix} \begin{pmatrix} v_1 \\ \vdots \\ v_n \end{pmatrix}$$

Since the gradient is zero at critical points, i.e., $\nabla \mathcal{L}(\theta_\mathbf{0}) = 0$, the quadratic approximation to the function around its critical points is given by: $\mathcal{L}(\theta) \approx \mathcal{L}(\theta_\mathbf{0}) + {}^1/{}_2 H \mathcal{L}(\theta - \theta_\mathbf{0})$. We see that the signature of the Hessian (the signs of its eigenvalues) precisely classifies the critical points ($\nabla \mathcal{L}(\theta_\mathbf{0}) = 0$) as a local maximum (all eigenvalues are negative), minimum (all eigenvalues are positive) or saddle (eigenvalues have mixed signs). Computing a local Hessian yields rich information about the local curvature around the optimal point in the neural network parameter space.

# 3 Methods

## 3.1 Diffusion Curvature

We now illustrate how diffusion can be used to measure the relative spreading of geodesics under the influence of Riemannian curvature. To build intuition, we first discuss the case of surfaces (dimension $n = 2$). Three canonical surfaces are the sphere, the cylinder, and the saddle. The Gaussian curvature of these surfaces are positive, zero, and negative, respectively. Imagine taking a sticker and trying to adhere it to one of these surfaces. The sticker, having been printed on a flat piece of paper, has zero Gaussian curvature. It will adhere perfectly to the cylinder, will bunch up (there will be too much sticker material) when trying to adhere it to the sphere, and will rip (there will be too little sticker material) when trying to adhere it to the saddle. This example showcases the area comparison definition [21, pp. 225–226] of Gaussian curvature, $k$, where one computes the limiting difference

between the area $A(r)$ of a geodesic disk on the manifold and a standard Euclidean disk,

$$k = \lim_{r \to 0^+} 12 \frac{\pi r^2 - A(r)}{\pi r^4}.$$

For example, Gaussian curvature will be positive when $\pi r^2 > A(r)$, i.e., when the area of the sticker exceeds the area of the corresponding geodesic disk on the sphere. The Bishop-Gromov volume comparison theorem formalizes how to extend this into dimensions $n > 2$. Here we will compare to $M_k^n$, the complete $n$-dimensional simply-connected Riemannian manifold of constant sectional curvature $k$, i.e the sphere of radius $1/\sqrt{k}$ when $k > 0$, Euclidean $n$-space when $k = 0$, and the appropriately scaled version of hyperbolic $n$-space when $k < 0$.

**Theorem 1** (Bishop-Gromov). *Let $(M^n, g)$ be a complete Riemannian manifold with Ricci curvature bounded below by $(n-1)k$. Let $B(p, r)$ denote the ball of radius $r$ about a point $p \in M$ and $\bar{B}(p_k, r)$ denote a ball of radius $r$ about a point $p_k \in M_k^n$ (as defined above). Then $\phi(r) = {}^{Vol(B(p,r))}/{}_{Vol(\bar{B}(p_k,r))}$ is a non-increasing function on $(0, \infty)$ which tends to $1$ as $r \to 0$. In particular, $Vol(B(p, r)) \leq Vol(\bar{B}(p_k, r))$.*

Theorem 1 captures the sticker phenomenon: as curvature increases, the volume of comparable geodesic balls decreases. Positive curvature corresponds to geodesic convergence and smaller volumes, whereas negative curvature corresponds to geodesic divergence (spread) and larger volumes. The discrete nature of the data manifold makes it impossible to compare volumes as the distance scale (radius) goes to zero. Theorem 2 gives us access to the intrinsic distance between data points, i.e., the diffusion distance, which we then use to define metric balls on the Riemannian manifold $M$.

**Theorem 2** (Coifman et al. [22]). *The diffusion map $\Phi_t(x_i) = [\lambda_1^t \phi_1(x_i), \ldots, \lambda_N^t \phi_N(x_i)]^T$ embeds data into a Euclidean space where the Euclidean distance is equal to the diffusion distance $D_m$, i.e. $D_m^2(x, y) = \|\Phi_t(x) - \Phi_t(y)\|^2 (1 + O(e^{-\alpha m}))$.*

We can now define $B_m(x, r) = \{y \in M : D_m(x, y) \leq r\} \subset M$ to be the ball centered at $x \in M$ with diffusion radius $r$. Let $B(x, r)$ denote the set of sampled points from $B_m(x, r)$ and let $|B(x, r)|$ denote the cardinality of this set. To define diffusion curvature, we generate a random walk from a point $x$ by diffusing a Dirac based at $x$, i.e., $m_x(\cdot) = \delta_x \mathbf{P}^t$ where $\delta_x$ is the one-hot vector, so that $m_x(y) = \mathbf{P}^t(x, y)$ is the transition probability from $x$ to $y$.

**Definition 1.** *The pointwise diffusion curvature $C(x)$ is the average probability that a random walk starting from a point $x$ ends within $B(x, r)$ after $t$ steps of data diffusion, i.e.,*

$$C(x) = \frac{\sum_{y \in B(x,r)} m_x(y)}{|B(x, r)|} \tag{2}$$

We can extend this definition to a contiguous region $U$ of the manifold $M$ consisting of neighboring points $U = \{x_j\}_{j=1}^k$ by defining $m_U(\cdot) = \delta_U \mathbf{P}^t$, where $\delta_U$ is the indicator vector on the set $U$.

We have $C(x) \in [0, 1/N]$ where $N = |B(x, r)|$ with larger values indicating higher curvature relative to lower values. The idea is to use diffusion probabilities to capture the relative spreading of geodesics. Intuitively, a random walker is more likely to return to their starting point in a region of positive curvature (where paths converge) than negative curvature (where paths diverge).

More precisely, in negatively curved regions, the random walker can get lost in the various divergent (disconnected) branches, whereas in positively curved regions the paths of the random walker exhibit more inter-connectivity, so that the return probability of a walk is higher. We make this idea more formal in the following:

**Theorem 3.** *If we sample uniformly from a Riemannian manifold $M^n$ with Ricci curvature bounded below by $k(n-1)$ then $C(x) \geq C(x_k)$, where $x \in M^n$ and $x_k \in M_k^n$ (as defined above).*

*Proof.* From Theorem 1 we have $vol(B_m(x, r)) \leq vol(\bar{B}_m(x_k, r))$ and together with uniform sampling this yields $|B(x, r)| \leq |\bar{B}(x_k, r)|$. We then have $1/|B(x, r)| \geq 1/|\bar{B}(x_k, r)|$ so that even with uniform transition probability measures we achieve the desired inequality. Moreover, as the probability measure $m_x(\cdot)$ is constructed by diffusing a Dirac according to affinities based on a Gaussian kernel, the measure decays with relative diffusion distance from $x$. Hence for sets centered

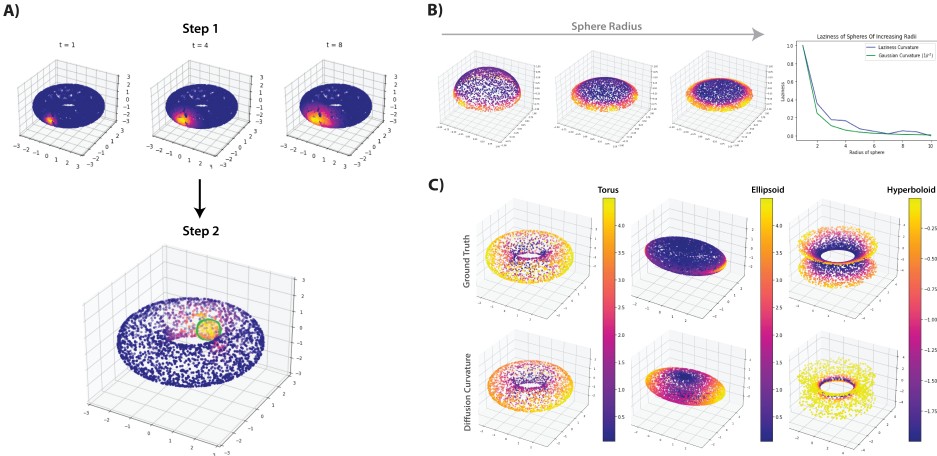

Figure 1: Diffusion recovers Gaussian curvature information (up to a scaling factor and offset) on test manifolds, e.g. sphere, torus, ellipsoid and hyperboloid.

at $x$, larger cardinality implies lower average transition probability. We may therefore conclude that

$$C(x) = \frac{\sum_{y \in B(x,r)} m_x(y)}{|B(x,r)|} \geq \frac{\sum_{y \in \bar{B}(x_k,r)} m_{x_k}(y)}{|\bar{B}(x_k,r)|} = C(x_k).$$

$\square$

**Limitations.** While diffusion curvature can be computed easily, it also necessitates an accurate sampling procedure, which can be problematic for difficult spaces. Moreover, due to its formulation, diffusion curvature only provides relative (non-negative) values.

### 3.2 Neural Network Implementation of Quadratic Forms

In the previous section, we have shown that the diffusion operator and associated diffusion probabilities contain information for computation of curvature from point cloud data. In this section, we extend this measure from a scalar quantity to an entire quadratic form by training a neural network on the diffusion map embedding of the point cloud. A schematic of our neural network, called CurveNet, is shown in Figure 2.

CurveNet takes as input points $\{X_i = (\mathbf{x}_i, y_i)\}_{i=1}^N$ sampled near a local minimum or saddle point $(\mathbf{x}_s, y_s)$. Here $\mathbf{x}_i \in \mathbb{R}^k$ are input parameters for an objective function or loss, $f$, with $y_i = f(\mathbf{x}_i)$ and $\nabla f(\mathbf{x}_s) = 0$. Points sampled from the parameter space of a neural network are mapped to an $N$ dimensional diffusion vector, $\Phi_t(X_i) = [\lambda_1^t \phi_1(X_i), \dots, \lambda_N^t \phi_n(X_i)]^T$. Recall from Equation 1 that the diffusion map is an eigendecomposition of the diffusion operator $\mathbf{P}^t$. Hence, without reduction of dimensionality, it encodes all information present in the diffusion operator, and contains $N$ dimensions for a sampling of $N$ points. Here we utilize this to estimate a $k \times k$ quadratic form of the sampled point cloud, $y_i = \mathbf{x}_i^T \hat{Q} \mathbf{x}_i$, using a neural network. Note that when the point cloud is a sample of a local region in the manifold, this gives a localized quadratic form describing 2nd order information in the region sampled.

Since our focus is on local quadratic approximation, we train CurveNet on samples from idealized surfaces of the form $f(x) = x^T Q x$ where $Q$ is a symmetric matrix with dimensionality dependent on the cardinality of the point cloud; see Figure 3 for some examples of generated 2-D surfaces. We generate a series of such surfaces by varying parameters within $Q$ and sampling points. We then provide the diffusion map embedding of the sampled points to the neural network and train it to predict $Q$ using a mean-squared error loss. We also include an $L_1$ regularization term to promote sparsity in the coefficients, which estimates the dimensionality of the input data, leading to

$$\mathcal{L}(Q, \hat{Q}) = \sum_{j=1}^k \sum_{j'=1}^j (Q_{jj'} - \hat{Q}_{jj'})^2 + \alpha |Q_{jj'}|. \tag{3}$$

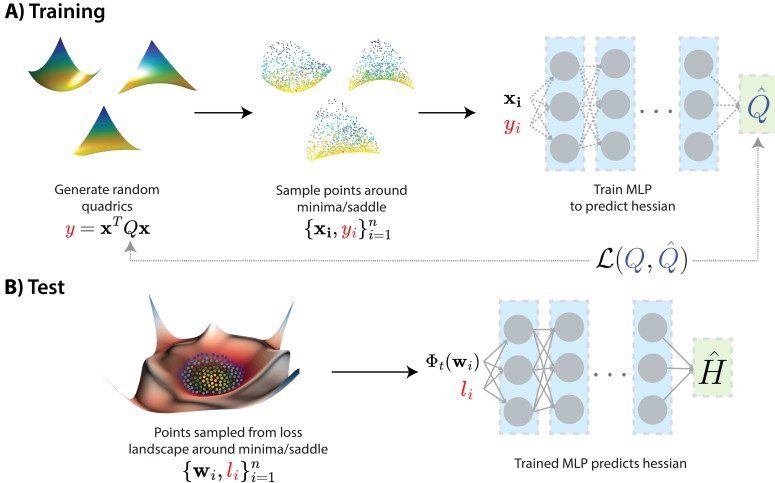

**A) Training**

Generate random
quadrics
$y = \mathbf{x}^T Q \mathbf{x}$

Sample points around
minima/saddle
$\{\mathbf{x_i}, y_i\}_{i=1}^n$

Train MLP
to predict hessian

$\mathbf{x_i}$
$y_i$

$\hat{Q}$

$\mathcal{L}(Q, \hat{Q})$

**B) Test**

Points sampled from loss
landscape around minima/saddle
$\{\mathbf{w}_i, l_i\}_{i=1}^n$

Trained MLP predicts hessian

$\Phi_t(\mathbf{w}_i)$
$l_i$

$\hat{H}$

Figure 2: The neural network is trained on points $(x_i, y_i)$ sampled from randomly generated quadrics and tested on weights $(W_i)$ and loss values $(l_i)$ sampled from the loss landscape of neural networks. Objective function values, $y_i$, and loss values, $l_i$, are treated as special axes containing curvature information. Points sampled from the loss landscape, $X_i$, are embedded using the diffusion map $\Phi_t$.

**Curvature and the Hessian.**    In the classical setting of surfaces, a relevant notion of curvature is the Gaussian curvature $\kappa$. In the special case that $M$ is realized as the graph of a function $f(x, y)$, the Gaussian curvature can be calculated explicitly from the Hessian matrix $H$ of $f$ by the formula

$$\kappa(x, y) = \frac{\det(H_{(x,y)}f)}{(1 + |\nabla_{(x,y)}f|^2)^2}.$$

Note that at a local optimum of $f$ (where $\nabla f = 0$) the Gaussian curvature is completely characterized by the eigenvalues of the Hessian. In higher dimensions, by contrast, a single number is no longer sufficient to capture the full curvature information of a Riemannian manifold, and one instead appeals to the Riemann curvature tensor. In the special case that $M$ is realized as the graph of a function $f \colon \mathbb{R}^n \to \mathbb{R}$, the spectrum of the Hessian of $f$ again contains important (albeit not complete) curvature information. As we are interested in measuring curvature in the setting of loss landscapes, we apply CurveNet described above to directly compute the Hessian of the parameters with respect to the loss function of a neural network in question.

## 4   Results

We validate both diffusion curvature and CurveNet using (a) synthetic test cases, and (b) single-cell RNA sequencing data (scRNA-seq). Moreover, we analyze the quality of our curvature estimation methods in a data sampling scenario, where we sample points in the vicinity of an optimum. All test cases are chosen to assess the efficiency and accuracy of our methods.

We trained CurveNet on $N = 1000$ samples from idealized quadratic surfaces. Later, we sampled 1000 points in the local neighborhood of the model parameters to estimate curvature of the loss landscape via the diffusion map embedding. As the parameter space of neural networks can be large, sampling points around the minima is a method of reducing dimensionality and restricting the estimation to a less complex space. In these local neighborhoods, the intrinsic dimension of the loss manifold is low, since trainable parameters are unlikely to change near the minimum. We used intrinsic dimensions of $k = \{2, 3, \ldots 20\}$ to generate idealized surfaces to ensure that CurveNet can approximate the Hessian within a reasonable range of intrinsic dimensions. We trained on 5000 different randomly generated quadrics, all sampled using $N = 1000$ points, for each intrinsic dimension. CurveNet then outputs a quadratic form with $K(K + 1)/2$ entries, where $(K = \max(k) = 20)$, which we compare with the known ground truth using mean squared error (Table 1). Training and testing were done on 8 core Tesla K80 GPUs with 24 GB memory/chip.

Table 1: MSE ($\mu \pm \sigma$) of neural network Hessian estimation. All values scaled by $10^3$, lower is better.

| Test Data
Intrinsic dim.
(# nonzero coeffs.) | CurveNet
(# epochs = 1000) | CurveNet
(# epochs = 100) | Baseline
(random) |
|---|---|---|---|
| 2 (4) | $2.115 \pm 1.250$ | $14.071 \pm 8.664$ | $612.145 \pm 2.836$ |
| 5 (16) | $0.964 \pm 0.144$ | $4.865 \pm 3.382$ | $554.652 \pm 1.704$ |
| 10 (56) | $0.752 \pm 0.013$ | $0.798 \pm 0.029$ | $530.203 \pm 0.645$ |
| 15 (121) | $0.766 \pm 0.022$ | $0.085 \pm 0.007$ | $521.050 \pm 0.485$ |
| 20 (211) | $0.712 \pm 0.034$ | $0.082 \pm 0.004$ | $515.739 \pm 0.397$ |

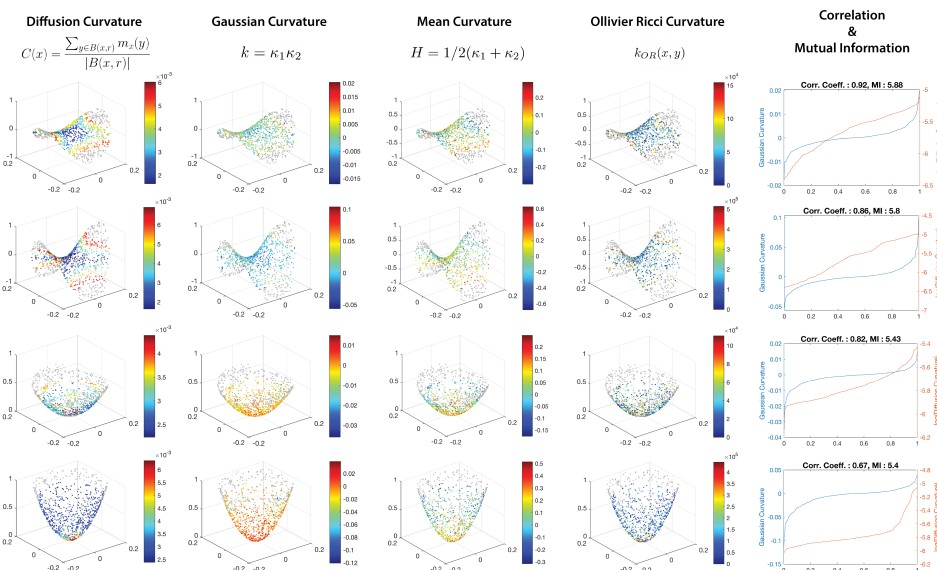

Figure 3: Curvature estimation on toy dataset of quadric surfaces.

## 4.1 Toy test cases for curvature estimation

First, we calculated diffusion curvature for simple 2-manifolds embedded in $\mathbb{R}^3$. We performed $t$-step random walks on points sampled from test surfaces like the torus, sphere, ellipsoid and hyperboloid, shown in Figure 1. The resulting diffusion curvature measurement is strongly correlated Gaussian curvature, a well known intrinsic notion of curvature. Next, we generated a series of synthetic datasets representing minima and saddles where the primary objective is to estimate curvature at central points (which are not affected by edge effects). Figure 3 shows a series of artificially generated $3D$ surfaces whose curvature varies in two principal directions from positive to negative. In the left column we show the curvature estimate given by diffusion curvature (Eqn. 2). The second, third and fourth columns show comparisons to Gaussian curvature (defined in Section 3.1), mean curvature and Ollivier's Ricci curvature (defined in Section 2.2) respectively. The last column contains biaxial plots showing the correlation between Gaussian and diffusion curvature.

We observe that diffusion curvature captures Gaussian Curvature on these data. Despite being scaled differently, diffusion curvature highlights essentially the same structures as Gaussian Curvature. This qualitative observation is quantified by the correlation. Overall, we obtain high correlations for the different surfaces. The last surface is slightly different, as high values measured by diffusion curvature are more concentrated within the "cusp", whereas Gaussian curvature spreads out such values over a larger part of the surface. This example demonstrates the benefits of diffusion curvature, though: in the context of loss landscape analysis, diffusion curvature is much more sensitive to such sharper minima, thus facilitating their detection.

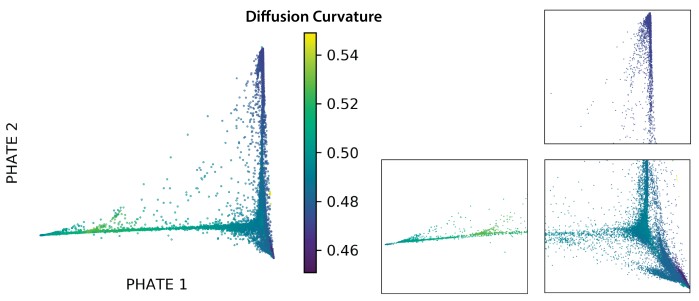

Figure 4: Diffusion curvature of the iPSC data.

## 4.2 Curvature estimation for single-cell data

We also estimated the curvature of a publicly available single-cell point cloud dataset obtained using mass cytometry of mouse fibroblasts. The dataset does not contain any identifiable or human information. Mass cytometry is used to quantitatively measure 2005 mouse fibroblast cells induced to undergo reprogramming into stem cell state using using 33 channels representing various protein biomarkers. Such a system is often called induced pluripotent stem cell (iPSC) reprogramming [23]. This dataset shows a progression of the fibroblasts to a point of divergence where two lineages emerge, one lineage which successfully reprograms and another lineage that undergoes apoptosis [4]. We note that our model correctly identifies the initial branching point (with cells that don't survive) as having low values of diffusion curvature indicating relatively negative curvature due to divergent paths out of the point (resulting in divergent random walks, see Figure 4). On the other hand it shows higher values indicating flat curvature along the horizontal branch. Diffusion curvature of another single-cell dataset, measuring embryonic stem cell differentiation, is provided in Section A1 of the Appendix.

## 4.3 Hessian estimation and sampling

To obtain second-order information around a critical point, $x_s$, we estimate a local Hessian: we first (for both toy test cases and neural networks) sample $n$ points, $x_1, x_2 \ldots x_n$, at random in the neighborhood of the critical point. Here $x_i \in \mathbb{R}^k$,, where $k$ is the full dimensionality of the domain of the toy functions or the neural network: $f : \mathbb{R}^k \to \mathbb{R}$ and the value of the objective function or loss, $f \in \mathbb{R}$ can be obtained by evaluating $f(x_i)$. In both settings, this allows us to obtain an $\mathbb{R}^{n \times (k+1)}$ matrix. Each row represents a sampled point $x_i \in \mathbb{R}^k$ with its associated loss or objective function in the last column. We consider this column as a special loss axis and use it as an input to the neural network to learn the coefficients along with the diffusion axes, which are obtained from the sampled points.

The diffusion axes were obtained by constructing a diffusion map in $\mathbb{R}^n$ based on the sampled points around the optimum, i.e., the minimum. We uniformly sampled 1000 points on a $k$-dimensional hypersphere which was then scaled by the parameter space of the optimum or saddle, as well as the gradient at that point. Special care was taken to ensure the points were sampled locally by evaluating at the relative difference between the evaluated loss at locally sampled points around the optimum or saddle and the actual loss at the optimum or saddle. We then use these diffusion coordinates $\Phi(x)$ and the value at the sampled point $f(x) \in \mathbb{R}$ as an input pair $(\Phi(x), f(x))$ to the neural network, which we ultimately use to estimate the Hessian. Figure 5 shows the eigenspectrum of the Hessian estimated using CurveNet for a feed-forward neural network trained to classify MNIST digits. We observe that the number of *negative* eigenvalues of the Hessian matrix (indicative of a maximum in the loss landscape) decrease over training epochs. In Figure 5, when comparing epoch 25 and epoch 200, we observe a marked shift in the (cumulative) density of eigenvalues towards positive eigenvalues, showing that the feed-forward neural network is approaching a minima in the parameter space. Hessian eigenspectrums estimated using CurveNet for a convolutional neural network and ResNet-18 are provided in the Appendix (Sections A2 and A3).

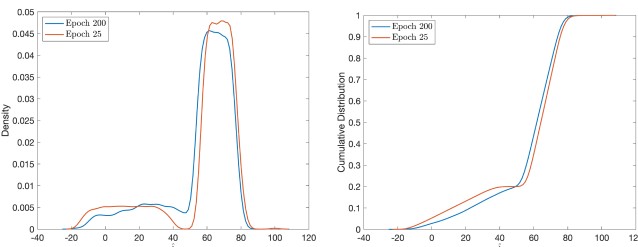

Figure 5: Density (left) and cumulative density (right) of the eigenspectrum when sampling around an optimum. At epoch 200, we observe a substantial decrease in the number of negative eigenvalues, showing that CurveNet is better capable of approximating the minimum.

## 5    Conclusion

We proposed *diffusion curvature*, a new measure of intrinsic local curvature of point clouds. Our measure leverages the laziness of random walks, obtained using the diffusion maps framework. We link diffusion curvature to existing volume comparison results from Riemannian geometry. In contrast to these notions, diffusion curvature can be computed effectively via neural networks even for high-dimensional data. While we demonstrated the effectiveness of such a curvature measure by analyzing numerous datasets of varying complexities, our formulation also leads to new research directions. Of particular interest will be proving additional results about our measure, relating it to existing quantities such as the Laplace–Beltrami operator, as well as formally proving its stability properties. We also want to develop new methods that use diffusion curvature to compare different datasets; being a quantity that is invariant under transformations (such as rotations of a point cloud), we consider diffusion curvature to be a suitable candidate for assessing the similarity of high-dimensional complex point clouds.

**Negative Impact:** We cannot envision any negative societal impact of this work.

## Acknowledgments and Disclosure of Funding

D.B. is funded by a Yale-Boehringer Ingelheim Biomedical Data Science Fellowship. S.K. received funding from the NIH (R01GM135929, R01GM130847, R01HD100035), NSF Career Grant (2047856), and the Sloan Fellowship (FG-2021-15883). D.B., K.M., D.T., B.R., I.A., and S.K. gratefully acknowledge funding from NSF DMS-2050398 which enabled this project.

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
