# A1 Diffusion curvature of embryonic stem cell differentiation

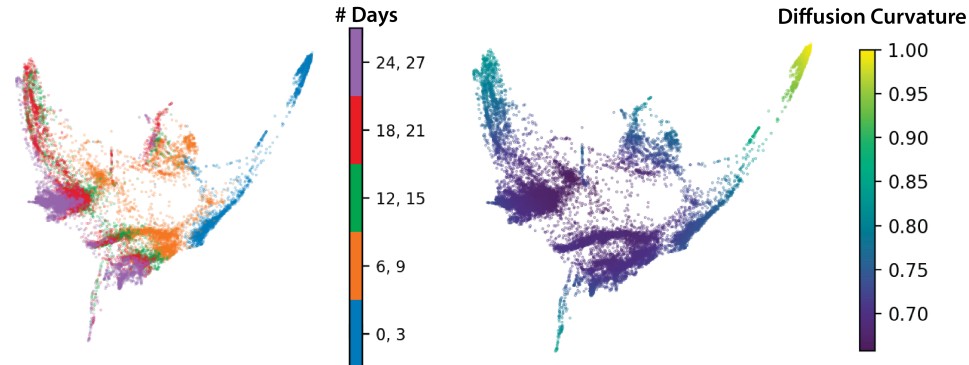

Figure A1: Diffusion curvature of embryonic stem cell differentiation. Left: PHATE visualization of scRNA-seq data color coded by time intervals. Right: PHATE plot colored by diffusion curvature values.

We applied diffusion curvature to a single-cell RNA-sequencing dataset of human embryonic stem cells [1]. These cells were grown as embryoid bodies over a period of 27 days, during which they start as human embryonic stem cells and differentiate into diverse cellular lineages including neural progenitors, cardiac progenitors, muscle progenitors, etc. This developmental process is visualized using PHATE in Figure A1 (left), where embryonic cells (at days 0-3, annotated in blue) progressively branch into the two large splits of endoderm (upper split) and ectoderm (lower split around day 6. Then during days 12-27 they differentiate in a tree-like manor into a multitude of lineages. Diffusion curvature, which is illustrated in the plot on the right shows that the tree-like structure that emerges during days 12-27 is consistently lower curvature than the initial trajectory which proceeds in a linear manner at days 0-3. This accords with the idea that divergent lineage structure is associated with low (negative) curvatures. Conversely, the endpoints of the transition corresponding to the stem cell state (days 0-3) and differentiated state (days 18-27) are associated with relatively high diffusion curvature values, indicative of positive curvature.

# A2 Hessian eigenspectrum of a convolutional neural net classifier

A CNN with 2 convolution layers ($5 \times 5$ kernel, stride length of 1, 2 pixels zero padding) containing 16 and 32 output channels respectively, ReLU activation, max pooling, and a fully connected layer, was trained to classify MNIST images. The model consists of $28,938$ trainable parameters, which were updated using the Adam optimizer with a learning rate of 0.01 during back-propagation. Parameter and loss values were saved at epochs 1, 2, 5, 10 and 12. The model was evaluated in the neighborhood of the saved parameters (at 1000 points sampled from $28,938$ dimensional hyperspheres of radii between 0.05 and 0.1, scaled by the parameter values) and its hessian was estimated using CurveNet. The eigenspectrum of the hessian (Figure A2) shows that the model parameters are at a saddle in epochs 1 and 2 as evidenced by the presence of both positive and negative eigenvalues. At epoch 5 or later, the absence of negative eigenvalues indicates that the parameters occupy a local minima, which increasingly becomes sharper with further training.

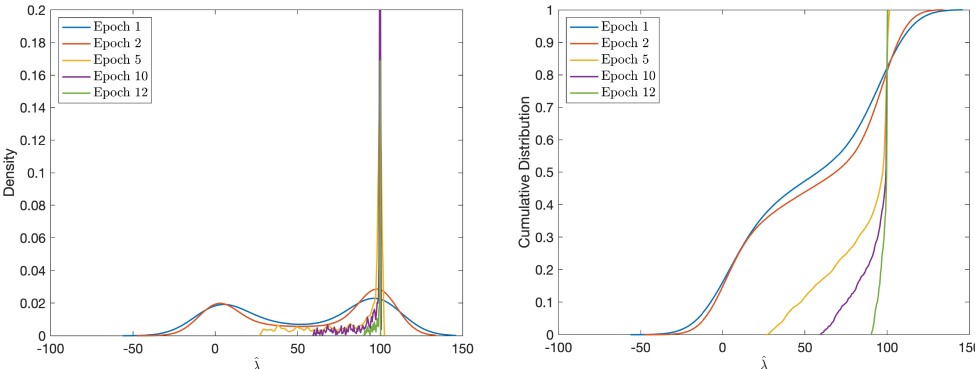

Figure A2: Density (left) and cumulative density (right) of the eigenspectrum when sampling around an optimum in the loss landscape of a CNN trained on MNIST. At epoch $\geq 5$, we do not observe any negative eigenvalues, indicating that the parameters have reached a local minima.

## A3 Hessian eigenspectrum of ResNet-18 classifier

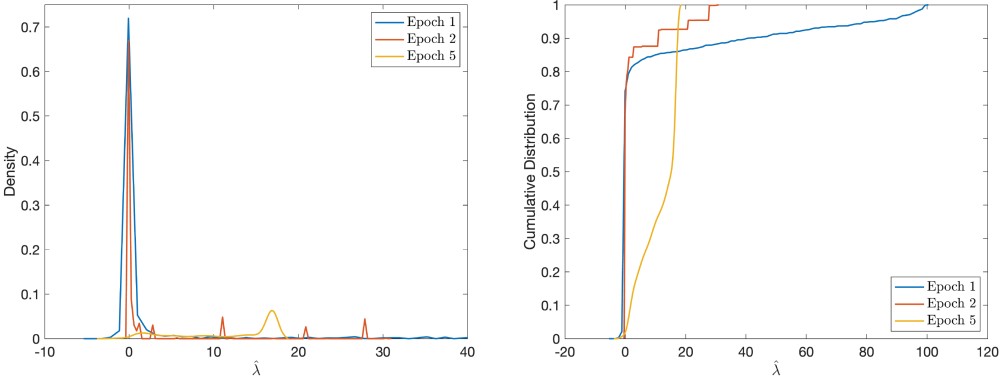

Figure A3: Density (left) and cumulative density (right) of the eigenspectrum when sampling around an optimum in the loss landscape of ResNet-18 trained on MNIST. At epoch $\geq 2$, we do not observe any negative eigenvalues, indicating that the parameters have reached a local minima.

We trained ResNet-18 from scratch on the MNIST dataset using the implementation provided by Torchvision. The default architecture was used, except the input layer was modified to accept single-channel, grayscale images from MNIST. The model consists of 11.2M parameters, which were trained using the RMSProp optimizer with a learning rate of 0.005. Parameter and loss values were saved at epochs 1, 2 and 5. The model was then evaluated at 1000 points sampled in the local neighborhood of the saved parameters, and its hessian was estimated using CurveNet. The eigenspectrum of the hessian (Figure A3) shows a relatively flat landscape at epoch 1 with most eigenvalues close to 0. Isolated peaks at positive eigenvalues appear at epoch 2, indicating that the parameter values have a reached a minima with loss increasing in a few directions and staying constant in others. At epoch 5, the parameter values have converged to a minima with large positive eigenvalues indicating that the loss increases in all directions of the parameter space.

## A4 Extrinsic curvature estimation using Gauss–Codazzi equations

A recent paper [2] proposes an orthogonal method to compute the *intrinsic* curvature of point-cloud data via *extrinsic* measurements. They compute the second fundamental form of the embedding which then yields the Riemann curvature tensor via the Gauss-Codazzi equation. Given orthonormal bases for the tangent space and normal space of the embedded manifold, the neighborhood of any point $p$ can be expressed in terms of coordinates $Y = [t_1, t_2 \ldots t_d, n_1, n_2 \ldots n_{n-d}]$ using regress functions $n_k = f_k(t_1, t_2 \ldots)$ to approximate the normal coordinates. The Riemann curvature tensor is recovered from the quadratic terms in the Taylor expansion of each $f_k$ via the Gauss-Codazzi equation. Scalar curvature is then computed by contracting the Riemannian curvature tensor. This method of computation becomes increasingly inefficient as the gap between ambient and intrinsic dimensions grows, as with scRNA-sequencing data. Moreover, similar underlying systems measured in different coordinates (i.e., single cell data measured in scATAC-seq and RNA-seq) can result in different *intrinsic* curvatures as this method is dependent on the embedding.

In Table A1, we compare the mean squared error (MSE) and runtime of the proposed CurveNet model against the second fundamental form scalar scurvature estimate using a test data of 100 randomly generated quadric hypersurfaces. Although the second fundamental form provide a better estimate of curvature, this methodology has a significantly higher computational expense in comparison to the trained neural network.

Table A1: Hessian estimation of randomly generated quadric hypersurfaces using CurveNet and Gauss–Codazzi equations. All MSE values scaled by $10^3$, lower is better. Average runtime in milliseconds is computed per test sample.

| Intrinsic dim. (# nonzero coeffs.) | CurveNet (1000 epochs) | | Gauss-Codazzi | |
|---|---|---|---|---|
| | MSE | Avg. Runtime (ms) | MSE | Avg. Runtime (ms) |
| 2 (4) | $2.115 \pm 1.250$ | 2.20 | $0.018 \pm 0.020$ | 2885 |
| 5 (16) | $0.964 \pm 0.144$ | 2.55 | $0.110 \pm 0.143$ | 2786 |
| 10 (56) | $0.752 \pm 0.013$ | 3.49 | $0.806 \pm 0.273$ | 5024 |
| 15 (121) | $0.766 \pm 0.022$ | 5.57 | $0.394 \pm 0.065$ | 14002 |
| 20 (211) | $0.712 \pm 0.034$ | 6.47 | $0.551 \pm 0.109$ | 56600 |

## A5 Relationship between diffusion operator and curvature

Given the strong relationship between diffusion coordinates and the eigenvectors of the Laplace–Beltrami operator, one can believe that these two objects capture similar properties of the manifold. As explored below, the eigenvalues of the Laplace–Beltrami operator relate to the curvature of a compact Riemannian manifold via the asymptotics of the trace of the heat kernel, and this lends credence to use of the diffusion probabilities to measure curvature on a data manifold. This section follows closely the exposition in [3].

On a compact Riemannian manifold $M$ we define the Laplace–Beltrami operator $\Delta$ to be $\Delta(f) = -div(grad(f))$. We consider the eigenvalues $\Delta(f) = \lambda f$ and note that the spectrum of $\Delta$ acting on $L^2(M)$ is a discrete set of non-negative numbers $\{0 = \lambda_0 < \lambda_1 \leq \lambda_2 \leq \cdots\}$ where each eigenvalue is written as many times as its multiplicity.

For compact Riemannian manifolds the heat kernel (or fundamental solution of the heat equation $u_t + \Delta(u) = 0$) exists uniquely and is given by

$$K(t, x, y) = \sum_j e^{-\lambda_j t} \phi_j(x) \phi_j(y)$$

where the $\lambda_j$ and $\phi_j$ are the eigenvalues and associated eigenfunctions of the Laplace–Beltrami operator, with the eigenfunctions normalized to form an orthonormal basis of $L^2(M)$.

We take the trace of the heat kernel and note that

$$Z(t) \equiv \int_M K(t, x, x) \, dx = \sum_{j=0}^{\infty} e^{-\lambda_j t}$$

so that the spectrum $\{\lambda_j\}$ determines the heat trace $Z(t)$.

Minakshisundaram and Pleijel's formula [4] for the asymptotic expansion of $Z(t)$ as $t \to 0^+$ is given by

$$Z(t) = (4\pi t)^{-n/2} \sum_{k=1}^{\infty} a_k t^k$$

where the coefficients $a_k$ are expressed via the metric and its derivatives. The first handful of coefficients have been calculated explicitly:

$$a_0 = vol(M), \qquad a_1 = \frac{1}{6} \int_M \tau, \qquad a_2 = \frac{1}{360} \int_M (5\tau^2 - 2|Ric|^2 - 10|R|^2)$$

where $n$ is the dimension, $\tau$ is the scalar curvature, $Ric$ is the Ricci tensor, and $R$ is the curvature tensor. In this way we see that the Laplace–Beltrami operator, via its eigenvalues, determines geometric information about the manifold, including the dimension, volume, and total scalar curvature. In the closed surface setting ($n = 2$) one can use Gauss-Bonnet to additionally recover the genus.

While the spectrum of the Laplace–Beltrami operator certainly contains interesting geometric information, it does not determine the isometry class of the manifold. Indeed, Milnor in 1964 produced the first examples of isospectral, non-isometric manifolds. Many additional examples have been produced, and with each pair one can reveal geometric information that is not encoded in the spectrum. These inaudible properties include the fundamental group, diameter, maximum scalar curvature, and even the local geometry of the manifold. One must therefore be prudent when using diffusion (and its relation to the Laplace–Beltrami operator) to prescribe curvature on a data manifold.

The Bochner formula similarly relates the Laplace-Beltrami operator on a Riemannian manifold to its Ricci tensor. It states

$$\frac{1}{2}\Delta|\nabla f|^2 = g(\nabla \Delta f, \nabla f) + |\nabla^2 f|^2 + Ric(\nabla f, \nabla f)$$

where $\Delta$ is the Laplace-Beltrami operator, $g(\cdot, \cdot)$ is the metric, $\nabla$ is the gradient, $\nabla^2$ is the Hessian, and $Ric$ is the Ricci tensor. When $f \in C^{\infty}(M)$ is harmonic, i.e. $\Delta(f) = 0$, the metric term disappears and we have

$$\frac{1}{2}\Delta|\nabla f|^2 = |\nabla^2 f|^2 + Ric(\nabla f, \nabla f).$$

We take these formulas as further evidence that the diffusion operator should contain information about the curvature of its associated data manifold.

# A6 Data and code availability

All source code and data are available under the MIT License at `https://github.com/KrishnaswamyLab/CurveNet`.