# OpenReview forum: "Diffusion Curvature for Estimating Local Curvature in High Dimensional Data"
_NeurIPS.cc/2022/Conference — NeurIPS 2022 Accept_

### Official Review · Reviewer_PnD7 · 2022-07-11

**Rating:** 6
**Confidence:** 4
**Soundness:** 3 good
**Presentation:** 3 good
**Contribution:** 4 excellent

**Summary:**

This paper introduces a new measure of the local curvature of a point cloud sampled from an underlying manifold: the so-called diffusion curvature. This measure uses the framework of diffusion maps and random walks. More precisely, the idea is to use random walks to capture the relative spreading of geodesics. Actually, a random walker is more likely to return to its starting point in a region where geodesic paths converge, i.e. in a region of positive curvature than negative curvature. The authors demonstrate by experiments that their notion of curvature scattering can be related to the Gaussian one.

Using a diffusion map, the authors also propose a way to locally approximate the Hessian associated with the loss function of a neural network for a given data set. This second-order approximation allows quantifying the (local) minima and thus quantifying their generalizability. Intuitively, the authors claim that flat minima generalize better than sharp minima. This estimation is done using a neural network: CurveNet.

The experiments are conducted on simulated data, which allows, in particular, the comparison with Gaussian curvature, and single-cell data.

**Questions:**

Here are some minor comments and suggestions:
* _l41:_ “the minima” Which one?  Which function?  Uniqueness?
* _l69:_ $\lVert.\rVert_1$ is not defined.
* _l77:_ Maybe add a sentence to justify why $\lambda_1=1$?
* _l88 (and the next times):_ Which annex?
* _Section 2.3_: Revise this paragraph to better motivate the introduction of the Hessian that arrives without context. Maybe write the paragraph the other way around and start talking about the generalizability of minima? Just a suggestion!
* _l129:_ No need to redefine the sphere.
* _l130:_ $K$ is not defined.
* _Theorem 1:_ The same notation ($B(p,r)$) is used to define two different objects.
* _Figure 1:_ The article does not refer to Figure 1.
* _l254:_ “sample $n$ points” According to which law?

**Limitations:**

This work is theoretical and has no short-term societal impact.

**Strengths And Weaknesses:**

I appreciated the efforts of the authors to give geometric insights, especially in dimension 2 (see section 3.1), throughout their paper. In general, the article is pleasant to read and well constructed.

In my opinion, one of the main limitations of this work concerns the sampling of points (also by the authors' admission, line 175). This point is not discussed here, and I wonder how this notion of diffusion curvature behaves in very high dimensions.

On the other hand, as much as the article is well worked, there is a lot of redrafting of supplementary material to be done. Several times in the article, the authors refer to it, but without specifying which part of the appendix. Moreover, the different sections are never put in context, and their titles are not explicit. I have the feeling that the authors have cut and pasted pieces of their articles to fit into the 9-pages limit without trying to recontextualize the different paragraphs. In the end, it makes this supplementary material difficult to follow.

---

> ### Author Response · Authors · 2022-08-02
> **Response to Reviewer PnD7**
>
> We thank the reviewer for the insightful comments and feedback.
>
> **Remark:** In my opinion, one of the main limitations of this work concerns the sampling of points (also by the authors' admission, line 175). This point is not discussed here, and I wonder how this notion of diffusion curvature behaves in very high dimensions.
>
> **Response:** Please see our response to Reviewer 4SQF, who raise a similar concern, quoted here:
>
> Our method explicitly focuses on the challenges posed by the high dimensionality of the weight space. The kernel used for computing diffusion curvature is inspired by manifold learning methods that preserve curvature information. We perform the Hessian estimation by sampling parameters from local patches near the minima. In these local neighborhoods, the intrinsic dimension of the loss manifold is low since a smaller subset of the weight parameters changes. Also, note that the grid-sampled weights near the minima/saddles do not have any curvature. Curvature is a joint function of the model weights (input) and the loss value (output), and there is no curvature without loss. We treat the loss as a special axis in our CurveNet architecture, and therefore curvature information does not get “washed out” in the embedding.
>
> **Remark:** On the other hand, as much as the article is well worked, there is a lot of redrafting of supplementary material to be done. Several times in the article, the authors refer to it, but without specifying which part of the appendix. Moreover, the different sections are never put in context, and their titles are not explicit.
>
> **Response:** We refer to specific sections of the Appendix in the revised main text. We have also modified the titles of sections of the Appendix to make them more explicit. All minor comments and suggested improvements (l41, l69, l77, l88, Section 2.3, l129, l130, Theorem 1, Figure 1, and l254) have been addressed in the revision.

---

### Official Review · Reviewer_39uS · 2022-07-11

**Rating:** 5
**Confidence:** 3
**Soundness:** 2 fair
**Presentation:** 2 fair
**Contribution:** 2 fair

**Summary:**

The paper presents a technique to estimate local curvatures on point clouds using diffusion maps. The proposed metric, diffusion curvature, is quadratic in nature and resembles the gaussian curvature, in terms of discriminatively. The authors demonstrate its usability for estimating local curvatures on high (and low)-dimensional data and hessians of a neural network loss. The underlying idea is simple: the diffusion curvature measures the divergence in a local neighbourhood wrt to a ball of a constant radius. It can be computed on a point cloud of any dimension.

**Questions:**

Please address the points raised in the weakness section.

**Limitations:**

yes

**Strengths And Weaknesses:**

Strengths:

1. The proposed diffusion curvature is simple yet sufficiently novel, to my best knowledge.
2. In general, the paper reads well. The main idea is made sufficiently clear.


Weakness:
1. As seen in figure 3, the proposed  diffusion curvature is more discriminative than gaussian one. Perhaps, it should also be compared with mean curvature as it is more discriminative than the gaussian one, especially in the flat regions. Is there any reason why comparison with mean curvature was ignored?
2. It is unclear how the diffusion curvature handle the noisy data. Given its discriminative power, it seems that diffusion curvature will be more affected by the noise in data as compared to the gaussian curvature. Repeating the hessian estimation (figure 3) with varied levels of noise in the data is important to understand how unstable the proposed metric is.
3. The experimental section is missing a practical use-case. A possible use-case could be the use of this metric to correct noisy 3D data obtained through various 3D sensors such as Kinect.

---

> ### Author Response · Authors · 2022-08-02
> **Response to Reviewer 39uS**
>
> We thank the reviewer for the insightful comments and feedback.
>
> **Remark:** As seen in figure 3, the proposed diffusion curvature is more discriminative than gaussian one. Perhaps, it should also be compared with mean curvature as it is more discriminative than the gaussian one, especially in the flat regions. Is there any reason why comparison with mean curvature was ignored?
>
> **Response:** We have added a comparison to mean curvature and Ollivier Ricci curvature in the revised manuscript (see Figure 3).
>
> **Remark:** It is unclear how the diffusion curvature handle the noisy data. Given its discriminative power, it seems that diffusion curvature will be more affected by the noise in data as compared to the gaussian curvature.
>
> **Response:** Diffusion-based dimensionality reduction and manifold learning can handle noisy data very well, as demonstrated through multiple publications applying these techniques to single-cell data, which is notoriously noisy (e.g. arXiv:1908.02831v1, DOI:10.1016/j.cell.2018.05.061, DOI: 10.1038/s41587-020-00803-5). Robustness to noise is an inherent property of these methods since they operate in graph Fourier space.
>
> **Remark:** The experimental section is missing a practical use-case.
>
> **Response:** We presented results from 2 different single-cell experiments in our manuscript. Diffusion curvature of induced pluripotent stem cell data is included in the main text (Section 4.2), and diffusion curvature of embryonic stem cell differentiation is included in the Appendix (Section A1). We estimated the eigenspectrum of the Hessian using our CurveNet method for a feed-forward neural net (main text, Section 4.3) and a convolutional neural network (Appendix, Section A2) trained on MNIST. In addition to the above, we have now included the eigenspectrum of the hessian of ResNet-18 in the revised manuscript (Appendix, Section A3).

---

### Official Review · Reviewer_4SQF · 2022-07-11

**Rating:** 3
**Confidence:** 5
**Soundness:** 1 poor
**Presentation:** 2 fair
**Contribution:** 1 poor

**Summary:**

The paper formulates a discretized scalar curvature for graph and point cloud data, using the diffusion rate of a random walk started at the query point. This is tested on toy cases, and on single-cell data, showing its basic efficacy and correlation with Gaussian curvature (in the toy cases).

Additionally, it considers a procedure for Hessian estimation using diffusion maps to perform dimensionality reduction. In particular, they first train a neural network to estimate Hessians from quadrics of various degrees. Next, given a local optimum, they tack on the label dimension and put it through a diffusion map for the embedding, and apply the neural network. This was tested on synthetic data from a high-dimensional hypersphere, giving again, some evidence of efficacy.

**Questions:**

1. How do you deal with the problem that the dimensions of your input data far outweigh your output label? This would seem to suggest that it would be easy for this information to get washed out in the embedding, and you might simply end up measuring curvature of the input.
2. It's repeatedly mentioned that sampling around the minima is a method of dimensionality reduction. I'm not sure I understand why this is the case. It seems to me that if you randomly perturb about a point in $R^{|W|}$ where $|W|$ is the dimension of the weight space, then you are still generating a random sample from a full-dimensional ball.
3. In practice, many diffusion maps methods are quite sensitive to the kernel choice when building a graph from a dataset. Did you find that your method was robust to changes in some of those parameters (e.g. $\alpha$)?
4. Did you test whether your estimated Hessians and their spectra actually gave information on how robust a local optimum was?

**Limitations:**

I think the method presents an interesting idea, but the exploration is lacking, and with a lack of strong experimental evidence, I'm not clear that its assumptions are valid. Most of my questions above reference my doubts and potential pitfalls I see for the method.

I would recommend that further work be done on experimental validation be done to address these concerns, if the paper is not accepted.

**Strengths And Weaknesses:**

Strengths:

- The method is certainly novel, and the Hessian optimization method tackles a problem of importance.
- I quite liked Figure 2, which succinctly summarized their Hessian estimation method.

Weaknesses:

- The scalar curvature quantity and the Hessian estimation method are only loosely related. Either one is not really necessary for the other. The linking concept is that of diffusion maps.
- The effectiveness of the Hessian optimization method is never tested on a real dataset, in the proposed application, as some measure of how well a local minimum might generalize (question 4 below).
- I think the method overlooks the challenges presented by the potential high dimensionality of the weight space. See question 1 below.
- The experimental comparisons are relatively weak, with no competing methods considered for either concept.
- There was no test for robustness to kernel parameters (see question 3).

---

> ### Author Response · Authors · 2022-08-02
> **Response to Reviewer 4SQF**
>
> **Remark:** The scalar curvature quantity and the Hessian estimation method are only loosely related. Either one is not really necessary for the other. The linking concept is that of diffusion maps.
>
> **Response:** In this paper, we establish a link between “diffusion curvature” and the Riemannian notion of curvature. We show that diffusion probabilities carry scalar curvature information, which is preserved under dimensionality reduction. Consequently, diffusion probabilities can be used to estimate the Hessian near a minima/saddle point where the gradient of the loss function vanishes. For a function of two variables, curvature $K$ = det(Hess($f$))/(1+$f_x^2$ + $f_y^2$)$^2$ = det(Hess($f$)) near a minima/saddle. Our CurveNet method takes diffusion probabilities as input and estimates the Hessian with high accuracy.
>
> **Remark:** The effectiveness of the Hessian optimization method is never tested on a real dataset, in the proposed application.
>
> **Response:** We tested CurveNet on parameters sampled from the loss landscapes of a feed-forward network (Section 4.3) and a CNN (Appendix, Section A2) trained on MNIST. In the revised manuscript, we also provide results for the ResNet-18 classifer (Appendix, Section A3), also trained on MNIST. Additionally, we report results from diffusion curvature estimation for two different single-cell datasets (Section 4.2 and Appendix Section A1).
>
> **Remark:** The experimental comparisons are relatively weak, with no competing methods considered for either concept.
>
> **Response:** We compare our diffusion curvature measure to Gaussian curvature in Figure 3. In the revised manuscript, we have also added comparisons to mean curvature and Ollivier Ricci Curvature. Hessian estimation using CurveNet is compared to an alternative method (Gauss-Codazzi) in Section A4 of the Appendix.
>
> **Remark:** There was no test for robustness to kernel parameters. In practice, many diffusion maps methods are quite sensitive to the kernel choice when building a graph from a dataset. Did you find that your method was robust to changes in some of those parameters (e.g. $\alpha$)?
>
> **Response:** We use an anisotropic kernel for computing diffusion curvature, and the parameters are chosen according to best practices. We refer the reader to existing literature (arXiv:1908.02831v1, DOI:10.1016/j.cell.2018.05.061, DOI: 10.1038/s41587-020-00803-5, etc.) that demonstrates robustness to kernel parameters.
>
> **Remark:** How do you deal with the problem that the dimensions of your input data far outweigh your output label? This would seem to suggest that it would be easy for this information to get washed out in the embedding, and you might simply end up measuring curvature of the input. It's repeatedly mentioned that sampling around the minima is a method of dimensionality reduction. I'm not sure I understand why this is the case. It seems to me that if you randomly perturb about a point in $R^{\|W\|}$ where $\|W\|$ is the dimension of the weight space, then you are still generating a random sample from a full-dimensional ball.
>
> **Response:** Our method explicitly focuses on the challenges posed by the high dimensionality of the weight space. The kernel used for computing diffusion curvature create a kernel inspired by manifold learning methods that preserve curvature information. We perform the Hessian estimation by sampling parameters from local patches near the minima. In these local neighborhoods, the intrinsic dimension of the loss manifold is low since a smaller subset of the weight parameters changes. Also, note that the grid-sampled weights near the minima/saddles do not have any curvature. Curvature is a joint function of the model weights (input) and the loss value (output), and there is no curvature without loss. We treat the loss as a special axis in our CurveNet architecture, and therefore curvature information does not get “washed out” in the embedding.
>
> **Remark:** Did you test whether your estimated Hessians and their spectra actually gave information on how robust a local optimum was?
>
> **Response:**  Changes in the spectra are consistent with improvements in train and test accuracy over epochs. Please see the discussion in Section 4.3 of the main text and Sections A2 and A3 in the Appendix.

---

> > ### Comment · Reviewer_4SQF · 2022-08-08
> > **Thank you and further thoughts**
> >
> > Thank you for the responses, which have clarified some points. They've also spurred a look back at the paper, and some additional reflections and concerns:
> >
> > 1. There seem to be some discrepancies between the Hessian estimation method description in Sections 3.2 and 4.3. In Section 3.2, it seems that the diffusion embedding is used to map both data points and their labels together: $X_i = (x_i, y_i)$. In Section 4.3, the diffusion embedding is only used on the data points. Which is it?
> >
> > 2. I think it would help clarity in Figure 2, if the training schematic also included the diffusion embedding. The fact that it is missing makes it seem as if CurvNet takes inputs of different dimensions in training and test phases.
> >
> > 3. It is unclear to me why any method would succeed when $k^2 >> N$, as would likely be the case with any reasonably-sized neural network. In this setting, there are more unknowns than equations, so it is impossible to determine the Hessian even with infinite compute resources. Indeed, Section 4.0 shows efficacy of CurvNet only in cases where $N > k^2$. To be convinced of efficacy, I'd be interested to see how it performs in the $k^2 >> N$ setting.
> >
> > Some responses to replies above:
> >
> > "...loosely related": I still find the link to be tenuous. The Hessian estimation method does not explicitly use diffusion curvature, but rather relies on the diffusion map to preserve geometry and curvature information. I have no argument with the suggested link, but I don't think it's strong enough to support inclusion of the definition and results for diffusion curvature with those for the Hessian estimation method, or vice versa.
> >
> > "...never tested on a real dataset": The results of Section 4.3 and those cited in the appendix assume that a neural network converges to better and better approximate local minima as training progresses. Most would agree that this is likely the case, but I don't think it is solid enough to serve as a proxy for showing that CurvNet is working. The stochasticity of the descent and the high dimensionality of the space mean that there's likely a high degree of noise. I would be more convinced if exact Hessians from a neural network were computed, and results from ConvNet were compared to that.
> >
> > "comparisons are relatively weak": When I speak of comparisons here, I was imagining comparisons to other methods that aim to determine quality of minima in neural networks, as this is the proposed practical application of the Hessian estimation. Quality of minima would need to be defined, but interesting characterizations like robustness to adversarial attack could be considered. Perhaps closer suggestions would be to compare to the works of Alain et al. and Sagun et al. which explicitly examine aspects of the Hessian eigenspectrum.
> >
> > "robustness to kernel parameters": It is simple to come up with scenarios where adjusting $\alpha$ too low or too high in the standard Gaussian kernel gives nonsensical results. At the least, some commentary on these kernel and parameter choices should be made in the text. Also, I would expect the use of an anisotropic kernel to be mentioned in the main text. Apologies if I missed it.
> >
> > "dimension concerns": Thanks for noting the lack of curvature from varying weights locally. I still find it surprising that a distance-based method (the kernels used are distance-based) would be able to pick out geometry or curvature distances from a single dimension out of many, unless the variation in the label/loss dimension is very high relative to variation in the data dimensions. Is this the case?
> >
> > Lastly, I still do not understand why the space of weights would be of lower dimensionality near a local minimum. In your experiment in Section 4.3, you sample from a $k$-dimentional hypersphere, considering the full dimensionality of the weight space.

---

> > > ### Author Response · Authors · 2022-08-09
> > > **Response to additional comments from Reviewer 4SQF**
> > >
> > > We thank the reviewer for the additional remarks and insightful critique.
> > >
> > > Discrepancy in the description of the Hessian estimation method in Sections 3.2 and 4.3: Thanks for bringing this to our attention. The diffusion embedding is only used for the data points. We have corrected Section 3.2 to reflect this.
> > >
> > > Diffusion embedding in Figure 2: The diffusion map embedding is only applied to weight parameters (not loss values) in the test phase. Please note that we do not use diffusion embedding during training, since CurveNet is independent of the coordinate system. We pad the coordinates of the points sampled from random quadrics with zeros during training to match the dimension of the diffusion map embedding (equal to number of points sampled, see Section 3.2) in the test phase.
> > >
> > > Model performance in the $k^2 >> N$ setting: In the appendix, we use CurveNet to predict the eigenspectrum of a ConvNet ($k$ = 28,938) and a ResNet ($k$ = 11.2M). In these cases, $N = 1000$, and $k^2 >> N$. Although the dimensionality of the weight space is very large, the intrinsic dimension near a minima/saddle is expected to be much lower, captured by diffusion map embedding. We note that the ground truth Hessian is not known for the ConvNet and ResNet architectures, so the efficacy of our method is not tested in this setting. We will perform experiments on synthetic data with known ground truth and include them in the next revision.
> > >
> > > "...loosely related": Although CurveNet does not use diffusion curvature explicitly, the link between diffusion probabilities and curvature, as established by our diffusion curvature result, provides a useful mathematical foundation for the presence of curvature information in the diffusion map embedding. Other reviewers have commended us for including diffusion curvature along with the relevant mathematical background. Since these results are related, we will reorganize the paper to present CurveNet and its applications as the main findings and move diffusion curvature to the appendix.
> > >
> > > "...never tested on a real dataset": Exact Hessian computation is intractable for neural networks of a reasonable size. We will include a comparison with the exact Hessian computation of a small neural network in the next revision.
> > >
> > > "comparisons are relatively weak": Determining the quality of the minima using CurveNet’s eigenspectrum prediction is the intended future direction of our work, and we agree that comparisons to other methods would be useful for showcasing the strengths of our proposed method.
> > >
> > > "robustness to kernel parameters": We agree that picking the right $\alpha$ is important. We have added a commentary on the choice of kernel and its parameters in the latest revision. In section 2.1.1 (first paragraph) we mention that the kernel we used is anisotropic.
> > >
> > > "dimension concerns": We estimate curvature using a distance-based kernel as the reviewer pointed out. Indeed the variation in the loss dimension is high compared to data dimensions, which should be investigated further. Please note that we extrinsically sample a $k$-dimensional hypersphere, however, the weight space has an *intrinsic dimensionality* lower than $k$ near the minima. Near a local minimum, the gradient is almost zero, so there are many parameters that have no influence on the loss. We are able to capture the interesting directions with a low-dimensional Hessian approximation.

---

### Official Review · Reviewer_r7PE · 2022-07-12

**Rating:** 5
**Confidence:** 4
**Soundness:** 3 good
**Presentation:** 3 good
**Contribution:** 2 fair

**Summary:**

This paper proposes a new local descriptor on sampled data, which quantifies the local recurrence properties of a random walk. This corresponds to a notion of curvature: random walks diffuse rapidly in regions that are negatively curved (such as branching trees), and stay more concentrated in regions of positive curvature (such as boundaries).


**Questions:**

Please refer to the questions above on implementation choices.

**Strengths And Weaknesses:**

This paper is original and well-written. It addresses an interesting topic in data sciences: what are the local descriptors that we can define (and understand!) on high-dimensional datasets.

Up to and including Section 3.1, this paper is truly excellent: the authors introduce their ideas clearly, with an appropriate level of details. I appreciate the fact that the authors included relevant references to concepts and theorems in the mathematical literature. An unfortunate trend at NeurIPS/ICML is to provide truncated overviews of the literature in pure mathematics in order to present classical ideas as “novel”: the diligent work of the authors should be rewarded.

Overall, I believe that this paper builds a very nice  bridge between modern geometry and machine learning applications. It introduces inspiring ideas in a way that is easy to digest by the NeurIPS community.

Please note, however, that I have some reservations about the content of pages 6 to 9:

- I do not understand why the authors decided to rely on a black-box neural network to estimate a local quadratic fit to the input point cloud. This method is intrinsically less reliable and interpretable than a simpler estimator, and goes against the clean mathematical definition of the proposed diffusion curvature. Couldn’t the authors have simply relied on a least-square formulation? Even if we add an L1 regularization term as in Eq. (3), it seems clear that we could compute a decent quadratic fit to the input point cloud by solving a small convex optimization problem with respect to Q. In 2022, this is easy to do using e.g. the “cvxpy" library.

- One of the main motivations that is proposed by the authors for their method is that it allows one to estimate the Hessian of a neural network. A discussion of alternative methods and approximations would have been appreciated.

- As far as I can tell, the authors do not discuss the scalability of their method. Since the proposed method relies on the computation of the first eigenvectors of a large N-by-N (normalized) kernel matrix, this is an important point. Could the authors explain why Eq. (1) is necessary? Couldn’t the authors simply iterate the application of a (normalized) Gaussian kernel matrix to simulate diffusion?
  I assume that the authors could be interested by recent papers and codes on diffusion or kernel matrices, such as "Fast geometric learning with symbolic matrices", Feydy et al., that was presented at NeurIPS 2020 (with a relevant example on eigendecompositions at: http://kernel-operations.io/keops/_auto_tutorials/backends/plot_scipy.html) or the work of Nicholas Sharp (https://nmwsharp.com/) on the vector Heat method and Diffusion networks.

- I would have appreciated some more details on the proposed experiments (e.g. better legends for the figures). Currently, it is hard to see them as anything more than toy examples: this could be fine for a theoretical paper, but there is certainly room for improvement here. For instance, the authors could illustrate the difference between their notion of curvature and other alternatives (discussed in Section 2.2) to highlight different behaviours, or discuss the influence of the choice of their kernel matrix (i.e. of the underlying random walk). This would be relevant for practitioners, in a context where the “best” local metric for a given dataset is often not known.

---

> ### Author Response · Authors · 2022-08-02
> **Response to Reviewer r7PE**
>
> We thank the reviewer for the insightful comments and feedback.
>
> **Remark:** I do not understand why the authors decided to rely on a black-box neural network to estimate a local quadratic fit to the input point cloud. This method is intrinsically less reliable and interpretable than a simpler estimator, and goes against the clean mathematical definition of the proposed diffusion curvature.
>
> **Response:** Indeed, the quadratic approximation $E[y_i] = \beta_{00} + \beta’x_i + x_i’Bx_i$ to the loss landscape at parameter values $x_i$ and loss $y_i$, where $\beta = (\beta_{10}, …, \beta_{k0})$ and $B = (\beta_{ij})$ for $i,j = 1, \cdots, k$ can be determined using least squares minimization of: $Q(\beta) = \sum_{i=1}^{N} (y_i - E[y_i])^2$ . In cases where $B$ is a symmetric matrix (e.g. Hessian), the quadratic coefficients $\beta_{ij}$ are restricted to a convex subset of $C_2^{(k+1)}$ - dimensional space, and therefore the estimation of $\beta_{ij}$ is a convex programming problem. The solution requires solving a system of $C_2^{(k+2)}$ linear normal equations. As the reviewer pointed out, this can easily be done using a package like “cvxpy” for small values of $k$. However, for deep neural networks with millions of parameters, this is much more computationally expensive. The neural network we trained to perform quadratic approximation can easily scale to problems of this size.
>
> **Remark:** One of the main motivations that is proposed by the authors for their method is that it allows one to estimate the Hessian of a neural network. A discussion of alternative methods and approximations would have been appreciated.
>
> **Response:** Few alternatives for estimating the full Hessian of neural networks are available. In the revised manuscript, we cite the work of Alain et al. (arXiv:1902.02366v1) on tracking the largest positive and largest negative eigenvalues of the Hessian. We also cite empirical results from Sagun et al. (arXiv:1706.04454v3), analyzing the spectrum of the Hessian with respect to the number of parameters (dimension) and data size. Related work on the visualization of the loss landscape by Li et al. (arXiv:1712.09913v3), is also cited.
>
> **Remark:** As far as I can tell, the authors do not discuss the scalability of their method. Since the proposed method relies on the computation of the first eigenvectors of a large N-by-N (normalized) kernel matrix, this is an important point.
>
> **Response:** In practice, only the first K eigenvalues of the N x N matrix (for some K << N) are of interest for assessing the generalizability and ease of training of large neural networks with millions of parameters ($N > 10^6$). In these cases, we can estimate the first K eigenvalues efficiently using power-based subspace tracking methods (Hua et al., DOI: 10.1006/dspr.1999.0348).
>
> **Remark:** Could the authors explain why Eq. (1) is necessary?
>
> **Response:** Eqn (1) in the paper defines the diffusion map. We refer to this definition in Theorem 2, where we state the result by Coifmann et al. that euclidean distances in the diffusion space are a good approximation to diffusion distances in the ambient space. We subsequently use this result to show that our definition of Diffusion Curvature respects the “sticker phenomenon” described by the Bishop-Gromov theorem in Riemannian geometry (proof of Theorem 3).
>
> **Remark:** I would have appreciated some more details on the proposed experiments (e.g. better legends for the figures). Currently, it is hard to see them as anything more than toy examples: this could be fine for a theoretical paper, but there is certainly room for improvement here.
>
> **Response:** We have provided more details and improved figure legends in the revised manuscript. In addition to “toy” examples like the sphere, torus, cylinder, and quadratic surfaces, we presented results from 2 different single-cell experiments in our manuscript. Diffusion curvature of Induced pluripotent stem cell data is included in the main text, and diffusion curvature of embryonic stem cell differentiation is included in the Appendix. Additionally, we estimated the eigenspectrum of the Hessian for feed-forward (main text) and convolutional neural networks (supplement) trained on MNIST. In addition to the above, we have now included the Hessian approximation of a ResNet network in the revised manuscript. We discussed an alternative notion of curvature, namely Ollivier’s Ricci Curvature (ORC), in Section 2.2. In the revised manuscript, we have included a comparison to ORC and mean curvature in Figure 3. Additionally, an alternative method for curvature computation via the Gauss-Codazzi equations is discussed in the Appendix (Section A4). Best practices for constructing the kernel matrix for diffusion random walks has been previously explored (arXiv:1908.02831v1, DOI:10.1016/j.cell.2018.05.061, DOI: 10.1038/s41587-020-00803-5, etc.).

---

### Meta-Review · Area_Chair_NS4K · 2022-08-25

**Recommendation:** Accept
**Confidence:** Less certain

**Metareview:**

This paper uses diffusion maps to measure curvature from point cloud data and includes some theoretical analysis as well as preliminary experiments demonstrating the value of the curvature measure.

The paper benefited from detailed discussion among a number of experts that gave it thorough consideration.  While the reviewers did not totally converge to a unanimous "accept" decision, the AC views their detailed/thoughtful discussion as a *positive* sign that the work will spark discussion and interest at the NeurIPS conference.  Other than limited experimental evaluation, it seems the main negative aspects of the work (mostly raised by reviewer 4SQF) are debatable in terms of whether they truly invalidate the research paper.

Overall, the AC recommends accepting this paper, especially since OpenReview will show the thoughtful discussion between the reviewers and authors.

**Award:**

No

---

### Decision · Program_Chairs · 2022-09-14

Accept